# Proteome Analysis of Molecular Events in Oral Pathogenesis and Virus: A Review with a Particular Focus on Periodontitis

**DOI:** 10.3390/ijms21155184

**Published:** 2020-07-22

**Authors:** Sachio Tsuchida

**Affiliations:** 1Division of Laboratory Medicine, Department of Pathology and Microbiology, Nihon University School of Medicine, Tokyo 173-8610, Japan; tsuchida.sachio@nihon-u.ac.jp; 2Division of Clinical Mass Spectrometry, Chiba University Hospital, Chiba 260-8677, Japan

**Keywords:** periodontal disease, gingival crevicular fluid, proteomics, MALDI-TOF MS, LC-MS

## Abstract

Some systemic diseases are unquestionably related to periodontal health, as periodontal disease can be an extension or manifestation of the primary disease process. One example is spontaneous gingival bleeding, resulting from anticoagulant treatment for cardiac diseases. One important aspect of periodontal therapy is the care of patients with poorly controlled disease who require surgery, such as patients with uncontrolled diabetes. We reviewed research on biomarkers and molecular events for various diseases, as well as candidate markers of periodontal disease. Content of this review: (1) Introduction, (2) Periodontal disease, (3) Bacterial and viral pathogens associated with periodontal disease, (4) Stem cells in periodontal tissue, (5) Clinical applications of mass spectrometry using MALDI-TOF-MS and LC-MS/MS-based proteomic analyses, (6) Proteome analysis of molecular events in oral pathogenesis of virus in GCF, saliva, and other oral Components in periodontal disease, (7) Outlook for the future and (8) Conclusions. This review discusses proteome analysis of molecular events in the pathogenesis of oral diseases and viruses, and has a particular focus on periodontitis.

## 1. Introduction

Periodontal disease is a generic term for diseases observed in periodontal tissue that consists of the gingiva, cementum, periodontal ligament, and alveolar bone, and are diseases that ultimately cause the loss of teeth and dysfunction in the oral cavity from the destruction of periodontal tissue due to disease progression [1,2,3]. The rate of disease progression is relatively slow, except when traumatic factors in the oral cavity accelerate the progression of lesions.

Although periodontal disease is said to be caused by certain bacteria, the type and number of bacteria do not distinguish the degree and progression of periodontal disease [4,5]. Proteomics can detect and quantify a number of proteins. Thus, it is possible to qualitatively and quantitatively evaluate how the process of synthesis of proteins in various normal tissues and tissues and cells with lesions changes before and after administration of a drug. By utilizing these characteristics, following the effects of drugs for periodontal disease and time-dependent changes in the disease state of periodontal disease, as well as identifying markers that change with the progress of periodontal disease, is considered feasible. Research on the presence of biomarkers and occurrence of molecular events is being conducted in the context of various diseases, and candidate markers of periodontal disease have been reported [6,7,8]. Increasing numbers of recent reports have also provided evidence that many approaches show promise for proteome analysis of molecular events in the oral pathogenesis of viral infections with a particular focus on periodontitis.

The purpose of this review was to investigate the recent progress in this field of proteome analysis.

## 2. Periodontal Disease

Periodontal disease originates in the gingiva and involves the periodontal tissues, the cementum, periodontal ligament, and alveolar bone, and the progression of gingivitis to periodontal disease is caused by irritation from the plaque lasting for a long period of time [9,10,11]. Worldwide research on periodontal and systemic diseases has resulted in a number of reports on the effects of general condition on periodontal tissues [12,13,14]. Both diabetes mellitus and periodontal disease are representative lifestyle habit illnesses and diet and smoking are associated lifestyle habit factors, with diabetes mellitus along with smoking said to be two big risk factors for periodontal disease [15,16,17,18]. Similarly, there are rare systemic disorders that are associated with severe periodontal disease, such as Papillon–Lefèvre syndrome [19]. Other systematic disorders, including neoplastic disease, may have an impact on the periodontium via mechanisms that are independent of those regulating the formation of dental plaque biofilm [20]. *Periodontal medicine* is a term used to describe the treatment of periodontal infection and inflammation [21]. Periodontitis has reported been linked to more than 50 systemic diseases. Research in periodontal medicine research over the past 100 years has focused in particular on the adverse effects of periodontal disease on two pathological conditions (cardiovascular disease and diabetes mellitus) and on pregnancy outcomes. The majority of cross-sectional, case-control, and longitudinal studies have revealed positive associations between poor periodontal status and the severity of cardiovascular disease, poor diabetes metabolic control, and adverse pregnancy outcomes. However, the detailed mechanism by which systemic disease leads to periodontal disease remains unclear. Although the onset of periodontal disease is reportedly triggered by bacterial infection, there are many unclear points in the mechanism of the disease state [21]. 

Dental plaque that forms on the teeth within minutes after cleaning is the primary cause of gingivitis. Plaque is composed of food particles and various types of bacteria that can adhere to teeth at and below the gum line. The bacteria present in dental plaque produce toxins that can irritate the gums.

The diagnosis of the “non-periodontitis” type of periodontitis is based on detailed clinical examinations, medical and dental history, tooth mobility, and radiographic assessments. The primary clinical evaluations performed to generate this diagnosis include probing pocket depth, determining the degree of bleeding on probing and clinical attachment loss, plaque index, and radiography. Probing depth and clinical attachment loss are determined using a UNC-15 periodontal probe placed at six specific sites (mesiobuccal, midbuccal, distobuccal, mesiopalatal/lingual, midpalatal/lingual, and distopalatal/lingual) on each tooth present in the oral cavity. Radiography will also provide critical information with respect to alveolar bone loss. 

A 2017 workshop developed a new classification framework for diagnosing periodontitis [22]. Substantial new information has emerged from clinical studies and basic science, as well as evidence from prospective studies designed to evaluate environmental and systemic risk factors for developing this disease. The workshop participants identified three distinct forms of periodontitis, including periodontitis, necrotizing periodontitis, and periodontitis as a manifestation of systemic disease [19]. The forms of the disease previously recognized as “chronic” or “aggressive”, are now grouped under the single heading of “periodontitis”, which is consistent with current knowledge on disease pathophysiology and pathogenesis. In revising these classifications, participants at the workshop agreed on a classification framework for periodontitis that was based on a multidimensional staging and grading system that can be adapted over time as new evidence emerges [23].

The diagnosis of periodontal disease is based on a detailed clinical examination, medical and dental histories, degree of mobility of a given tooth, and the radiographic assessment. The probing pocket depth, bleeding on probing, clinical attachment loss, plaque index, and radiography findings currently used to diagnose periodontal disease are in fact indicators of previous disease rather than present disease activity; as such, they have limited utility with respect to management and prognosis. Nonetheless, these clinical parameters provide important information about the current level of periodontal tissue destruction. The current classification of periodontal disease includes four stages which are based on disease severity and complexity of management. These categories include Stage 1 (initial periodontitis), Stage 2 (moderate periodontitis), Stage 3 (severe periodontitis with potential for additional tooth loss), and Stage 4 (severe periodontitis with potential for loss of dentition). Periodontal disease is also classified by its extent and distribution, including localized, generalized, or as having a molar-incisor distribution. Finally, the disease is categorized by grade based on evidence of risk of rapid progression and anticipated treatment responses, including Grade A (slow rate of progression), Grade B (moderate rate of progression), and Grade C (rapid rate of progression). 

Periodontitis therapy is initial nonsurgical therapy for the removal of plaque and tartar by scaling and root planning. In conventional periodontal treatment, mechanical tools such as hand curettes and power-driven scalers are used mainly for subgingival debridement in periodontal pockets to arrest the disease.

Flap surgery is the preferred treatment for advanced periodontal disease [24,25,26]. It is a simple procedure because it is an in-depth cleaning procedure that requires only local anesthesia. The stages of more advanced periodontal disease are characterized by deep periodontal pockets around the teeth that harbor bacteria, plaque, and tartar buildup and cause the destruction of the surrounding bone and soft tissues. Osseous surgery involves the cutting and placing of “flaps” of gingival tissue to provide access for thorough cleaning of the tooth. In cases where the periodontal disease has affected the alveolar bone, it is reshaped to promote the reattachment of the gingiva.

Recently, various kinds of lasers have been applied in dentistry. Erbium-doped yittrium-aluminum-garnet (Er:YAG) lasers have been used for procedures involving both soft and hard tissue. The high-level Er:YAG laser can provide low-level laser treatment (LLLT) around tissue. This effect has been proposed by Ohshiro et al. [27]. In clinical practice, LLLT is reported to stimulate wound healing, reduce inflammation, and alleviate pain after surgery. Additionally, according to many reports, a large variety of lasers exert biostimulation on periodontal cells in vitro. We previously reported the changes in protein expression of human gingival fibroblasts (HGFs) induced by low-level Er:YAG laser irradiation as studied in gel-free proteomic analysis, which provided information about the mechanisms of photobiomodulatory effects [28].

## 3. Bacterial and Viral Pathogens Associated with Periodontal Disease

Bacterial pathogens have been strongly associated with the onset and progression of periodontal disease. A consideration of some of the more general aspects of microbial dental plaques, including heterogeneity, development, microbial succession, composition, structure, and mechanisms of formation, necessarily precedes a discussion of the microbial composition of the gingival crevice (i.e., subgingival plaque). Microbial dental plaques are frequently heterogeneous, dense, non-calcified bacterial masses that are intimately associated with the tooth surface and are generally so firmly adherent that they cannot be washed off by salivary flow. By contrast, some forms of subgingival plaque are nonadherent and include large numbers of motile organisms.

Socransky et al. [29] performed DNA hybridization on more than 13,000 subgingival plaque samples from 185 adult subjects and confirmed the presence of specific microbial groups within dental plaques by cluster analysis and community ranking techniques. Most of the bacterial species identified were recognized as periodontal pathogens from the “red” and “orange” complexes, including *Porphyromonas gingivalis*, *Tannerella forsythia*, or *Parvimonas micra*, together with species from the genera *Fusobacterium*, *Prevotella* and *Treponema* [30,31,32,33,34,35]. Studies using “open-ended” molecular techniques [36,37,38] presented a higher microbial diversity and identified several uncommon taxa, including *Filifactor alocis, Enterococcus faecalis*, and species from the genera *Desulfobulbus*, *Dialister*, *Fretibacterium*, and *Rothia*. Interestingly, most of these species and genera have been associated with chronic or aggressive forms of periodontitis [39,40].

Two complexes consisting of microorganisms are believed to be the major etiological agents of periodontal disease. One complex, known as the “Red Complex”, consists of the following three closely related species: *Porphyromonas gingivalis*, *Tannerella forsythia*, and *Treponema denticola*. The second one, the “Orange Complex”, consists of *Fusobacterium nucleatum*, *Prevotella intermedia*, *Prevotella nigrescens*, *Peptostreptococcus micros*, *Streptococcus constellatus*, *Eubacterium nodatum*, *Campylobacter showae*, *Campylobacter gracilis*, and *Campylobacter rectus*. The development and progression of periodontal disease have been associated with the presence of specific Gram-negative bacteria, such as *Tannerella forsythia*, *Treponema denticola*, and *Porphyromonas gingivalis,* in plaque [41,42]. Persistence of *P. gingivalis* in subgingival plaque after oral treatment for periodontal disease is associated with progressive alveolar bone loss. Moreover, *P. gingivalis* plays a role in systemic conditions including atherosclerotic heart disease and aspiration pneumonia [43].

Viruses may participate in the pathogenesis of periodontitis by altering immunological defenses, by activating destructive host reactions, or through direct lytic effects on periodontal tissues [44,45]. One study demonstrated that the prevalence of human cytomegalovirus and Epstein–Barr virus (EBV) in subgingival specimens from adult patients with periodontitis was significantly higher than that observed in similar samples from healthy patients or those with gingivitis alone [46]. Furthermore, studies have demonstrated a high prevalence and high copy counts of cytomegalovirus, EBV, and herpes simplex virus type 1 in patients with progressive periodontal disease [47,48,49,50,51]. There is also evidence of a periodontopathic role of herpesviruses, according to association studies and immunology-based research; however, the specific molecular mechanisms through which herpesviruses may cause or exacerbate periodontitis have not yet been identified. Periodontal herpesviruses that disseminate via the systemic circulation to nonoral sites may represent a major link between periodontitis and systemic diseases [52]. Current treatment of periodontitis focuses almost exclusively on bacterial biofilm; periodontal therapy that targets both herpesviruses and bacterial pathogens can provide long-term clinical improvement and potentially reduces the risk of systemic diseases. Molecular diagnostic tests for periodontal pathogens may enable early microbial identification and preemptive therapy [52].

## 4. Stem Cells in Periodontal Tissue

Stem cells can exert both immunomodulatory and pro-osteogenic activities in the local environment. There has been increasing research interest in stem cells from the scientific community, primarily because of their ability to regenerate damaged tissues and also their potential in modulating inflammatory and immune responses. Dental pulp stem cells (DPSCs) are the most investigated mesenchymal stem cells (MSCs) from dental tissues; however, the oral cavity hosts several other stem cell lineages that have also been reported to be a better alternative in bone tissue engineering [53,54]. 

Currently, mesenchymal stem cells (MSCs) are being evaluated in preclinical and clinical trials for their ability to support wound healing and tissue regeneration [55]. MSCs are known to exhibit a therapeutic potential that is largely dependent on their ability to secrete proregenerative cytokines, thus rendering these cells an attractive option for improving the treatment of chronic wounds. The wound microenvironment is a miscellaneous key factor in the local management of the healing process; players such as the extracellular matrix or the resident and recruited cells with paracrine activity can determine the way and the appropriateness of the regenerative processes [54]. The most reported translational use of MSC/D-dMSC therapy is related to bone tissue regeneration; in fact, several authors have investigated the osteogenic ability of different stem cell types and genes, such as TGF-β1, that enhance the commitment of MSCs toward either the osteogenic or adipogenic lineages by reorganizing the actin cytoskeleton [56], as well as the use of a platelet-rich plasma blood clot stabilizer to treat infrabony periodontal defects [57]. 

Marrazzo et al. reported about the “safe” in vitro reparative models that work without any additive to safely apply such protocols in human models [58]. They focused on the effects of human platelet lysate (PL) on the repairing properties of dental pulp stem cells (DPSCs) by performing in vitro migration and survival assays. Their results suggested that human allogenic PL is a suitable alternative to fetal bovine serum (FBS) that can be used for the expansion and differentiation of DPSCs in vitro [58]. Their study is the first to report the influence of PL on DPSC migration and the antioxidant effects during ex vivo maintenance [58]. The use of PL in cell cultures did not impair the cell surface signature typically expressed by MSCs and even upregulated the transcription of Sox2 (a transcription factor critical for the maintenance of pluripotency and neurogenesis). Interestingly, DPSCs cultured in the presence of PL exhibited a higher healing rate after injury and were less susceptible to toxicity mediated by exogenous H_2_O_2_ than those cultured in the presence of FBS. Furthermore, PL addition was demonstrated as a suitable option for protocols promoting the osteogenic and chondrogenic differentiation of DPSCs [58].

## 5. Clinical Applications of Mass Spectrometry Using MALDI-TOF-MS and LC-MS/MS–Based Proteomic Analyses

Mass spectrometry (MS) is a powerful analytical tool with which researchers measure the mass-to-charge ratio of one or more molecules in a sample. In comparison with conventional methods, such as immunoassays, MS can help detect multiple analytical targets of interest simultaneously with improved specificity. MS technology has become significantly robust and sophisticated and, as a result, has been increasingly adopted in various subdisciplines of laboratory medicine (Table 1). Identification of peptides and proteins or of proteomic differential displays in body fluids and other clinical samples has applications in proteomics [59,60,61]. Technical developments in proteomics have progressed for the purpose of identifying all proteins expressed in a cell line, tissue, or body. Two-dimensional gel electrophoresis can reveal many proteins present in cells and tissues at any given time [62,63,64].

Numerous diseases, including periodontal diseases, have a profound effect on normal metabolic activity in the human body because they alter the expression of proteins in cells. Detailed protein information in vivo must therefore be detected and analyzed in order to understand the progression of disease, to determine the cause, and to develop new therapies. In other words, the information from the periodontal tissue lesion must be understood within the context of periodontal disease.

To identify proteins extracted from cells, tissues, and body fluids, cells are placed in one-dimensional or two-dimensional electrophoresis gel, and the proteins are extracted and subjected to matrix-assisted laser desorption/ionization time-of-flight mass spectrometry (MALDI-TOF MS). After the biological samples are digested by enzymes such as trypsin, the amounts of cleaved peptides are measured by liquid chromatography-mass spectrometry (LC-MS), and various proteins in tissues and organs can be analyzed. LC-MS is another essential tool in screening for inborn errors of metabolism, for toxicology analyses, and for therapeutic drug monitoring. Moreover, although the mainstream method for measuring corticosteroid and peptide hormones is various immunoassays, the use of liquid chromatography with tandem mass spectrometry (LC-MS/MS) is rapidly spreading. In addition, various MS techniques are becoming popular in basic research for periodontal disease.

As one of the key concepts in the post-genome era, the proteome has been investigated, and disease-marker searches with gel-based and gel-free proteome analysis techniques are being performed widely. The results of proteome research have attracted attention because they are connected directly with diagnostic application and development of therapeutic targets [62,63,64].

“Shotgun proteomics” involves the separation of complex peptide mixtures, which are generated by proteases such as trypsin, subjected to tandem MS, and analyzed with automated database searches [65,66,67,68]. Proteome analysis involves sample preparation, peptide digestion, protein identification, and either functional classification or comparative analyses. Shotgun proteomics, developed in recent years for high-throughput proteomic analyses [69,70,71,72,73], involves the use of bottom-up proteomics methods in which proteins in a biological sample are digested with proteases and then analyzed with MS. Quantitative proteomic methods have been the standard for analyzing proteins in cells and many tissues, as well as for identifying novel biomarkers.

Two systems and their associated databases, notably, the MALDI Biotyper (Bruker Daltonics, Bremen, Germany) and the VITEK^®^ MS (bioMerieux, Marcy I’Etoile, France), are currently in wide use for MALDI-TOF MS-based bacterial identification (ID). Although the analytical principles of the two systems are similar to one another, there are differences with respect to the databases and the construction of the diagnostic algorithms. Furthermore, the modes of data presentation remain somewhat different. For example, in the MALDI Biotyper system, the results of a pattern-matching process are presented as proposed by the manufacturer, with scores ranging from 0 to 3; scores below 1.7 are regarded as unreliable; scores between 1.7 and 2.0 are genus-level IDs and scores >2.0 can be evaluated at the species level. The Vitex^®^ MS system generates a confidence score that is presented as a percent probability. The percentage probabilities for a correct ID range from 60 to 99; values closer to 99 are those indicating a closer match. Percent probabilities under 60 are considered as unidentified. For both systems, however, lowering the cut-off values for bacterial ID from specimens from blood culture bottles increases the sensitivity of the test without compromising the specificity [74,75]. 

## 6. Proteome Analysis of Molecular Events in Oral Pathogenesis of Virus in GCF, Saliva, and Other Oral Components in Periodontal Disease 

### 6.1. GCF and Saliva

Currently, there are numerous substances produced during periodontal tissue remodeling as reported in basic research in periodontal diseases; these substances best reflect the condition of the periodontal tissue and are so-called proximal fluids near lesion areas, also referred to as gingival crevicular fluid (GCF) that are widely used as samples [76,77,78]. The gingival sulcus is the furrow between the teeth and the gums, and the gingival sulcus exists around the teeth in a V-shaped space. The tissue fluid exuding from the gingival sulcus is GCF. In general, GCF is collected by inserting a paper point under the gingival margin after simple exclusion of moisture.

Numerous enzymes and proteins related to the metabolism of periodontal tissue exist in GCF, and it is said to be a very important index for understanding the progression of periodontitis and evaluating the disease status. However, the amount of enzymes and proteins present in GCF is extremely low. The use of mass spectrometry is considered to be applicable to the quantitative analysis of minute amounts of proteins in periodontal tissue or GCF [79,80,81]. The various proteins involved in the progression of periodontal disease and destruction of periodontal tissue, such as enzymes that express or increase specific proteins in each disease state of periodontal disease or result from cell destruction, are present in GCF. Among these, candidate protein markers of periodontal disease are expected to exist. 

Many proteins detected in gingival crevicular fluid (GCF) play central roles in periodontal tissue turnover. Analysis of biochemical markers in GCF may therefore be useful for the diagnosis and monitoring of periodontal disease progression. In contrast, salivary biomarkers can allow for overall assessment of the disease status, as opposed to site-specific assessment facilitated by GCF analysis. Giannobile et al. and Ramseier et al. reported the use of salivary biomarkers related to periodontal disease as a diagnostic tool [82,83]. Analysis of saliva may be a useful screening tool for periodontal disease and for monitoring the responses to treatment [82,83]. Diagnostic biomarkers in the saliva of those with periodontal disease include proteins of host origin, host cells, bacteria, and bacterial products; saliva contains both host- and microbe-derived factors, including several protein-degrading enzymes, proteoglycans, lipids, and carbohydrates [84].

Ngo et al. used MALDI-MS and LC MALDI-MS of undigested GCF together with LC-MS of in-gel digested proteins to determine the peptide/protein composition of GCF [85]. Using these techniques, they identified 33 peptides, corresponding to cleavage products that had not previously been reported in GCF, and 66 proteins, including 43 that were newly identified; their research represents the most comprehensive proteomic study of GCF to date. Their use of reversed-phase high-performance liquid chromatography (RP-HPLC) followed by MALDI-TOF/TOF MS/MS was a new method of identifying GCF peptides. Moreover, Ngo et al. assessed whether the mass spectrometric analysis of GCF allowed for the site-specific prediction of periodontal disease progression [86]. Forty-one patients undergoing periodontal maintenance were monitored for 12 months; clinical measurements were taken at baseline and every 3 months thereafter. MALDI-TOF MS was used to analyze GCF according to pocket depth, modified gingival index, plaque index, and attachment loss. A genetic algorithm was used to create a model based on pattern analysis to predict which sites would undergo attachment loss. The clinical indices pocket depth, modified gingival index, plaque levels, and bleeding on probing were poor discriminators of mass spectra of GCF. Models generated from the GCF mass spectra could predict attachment loss at a site with high specificity, 97% recognition capability, and 67% cross-validation [86].

Chaiyarit et al. determined the potential use of MALDI-TOF/TOF MS for analyzing specific patterns of mass signals of low-molecular-weight proteins in saliva from patients with different oral diseases [87]. They collected unstimulated whole saliva from healthy subjects and patients with oral diseases, including oral cancer, oral lichen planus, and chronic periodontitis. They used MALDI-TOF/TOF MS to evaluate proteomic profiles of 5000- to 15,000-Da salivary proteins. The percentages of mass signals at 5592.26 and 8301.46 Da from oral cancer were significantly higher than those of the other diseases (*p* = 0.002 and *p* = 0.030, respectively). In oral lichen planus, the percentages of mass signals at 12,964.55 and 13,279.08 Da were significantly than those of the other diseases (*p* < 0.001). In chronic periodontitis, the percentages of mass signals at 5835.73 and 9801.83 Da were significantly lower than those of the other diseases (*p* = 0.003 and *p* = 0.005, respectively) [87].

Antezack et al. hypothesized that rapid routine and blinded MALDI-TOF analysis could accurately classify these three types of samples according to periodontal state [88]. Diagnostic tests based on protein profiles in saliva, GCF, and dental plaque were developed for the first time. Among 67 patients with periodontitis and 74 healthy controls, the decision trees enabled the diagnosis of periodontitis with 70.3% sensitivity (±0.211) and 77.8% specificity (±0.165) for saliva, 79.6% sensitivity (±0.188) and 75.7% specificity (±0.195) for GCF, and 72.1% sensitivity (±0.202) and 72.2% specificity (±0.195) for dental plaque. Both the sensitivity and specificity of the tests were improved to 100% (95% confidence intervals = 0.91–1 and 0.92–1, respectively) when two samples were tested [88].

Tang et al. studied salivary peptide biomarkers for screening patients with periodontal diseases by mass spectrometry [89]. Whole saliva, GCF, and serum samples were collected from 17 patients with chronic periodontitis, 17 with gingivitis, and 16 periodontally healthy persons who served as controls, and these samples were analyzed with MALDI-TOF MS. Nano liquid chromatography/electrospray ionization tandem mass spectrometry (nano-LC/ESI-MS/MS) was performed to identify possible proteins from which these peptides might be derived. Tang et al. reported that levels of most of the differentially expressed peptides were higher in participants with chronic periodontitis and gingivitis than in healthy controls [89]. Cluster analysis showed good differentiation between patients with chronic periodontitis and healthy controls. Most areas under the curve (AUCs) for differentially expressed peptides were >0.7, whereas some peptides from GCF and serum exhibited AUCs as high as 0.9–1.0 [89].

Bostanci et al. performed analysis of GCF exudatome from healthy and periodontally diseased sites by using LC-MS extraction, a label-free mass spectrometry method in which proteins can be simultaneously identified and quantified absolutely in biological fluids [90]. A total of 154 proteins of human, bacterial, and viral origin were identified in 40 GCF samples obtained from five healthy patients and five patients with generalized aggressive periodontitis. The proportion of viral protein was higher in the GCF samples from the patients with periodontal disease than in those from healthy patients. Bostanci et al. showed that higher levels of viral proteins (such as herpesvirus protein 2) in the diseased samples corroborated evidence from previous reports that confirmed the involvement of viral infection in the pathogenesis of periodontal disease [90].

### 6.2. Oral Disease Pathogenesis

Development and progression of periodontal disease are associated with the presence of specific Gram-negative bacteria, such as *Tannerella forsythia*, *Treponema denticola*, and *Porphyromonas gingivalis*, in the subgingival plaque [91]. The persistence of *P. gingivalis* in the subgingival plaque even after treatment for periodontal disease is significantly associated with progressive alveolar bone loss. Moreover, *P. gingivalis* has been implicated as an accessory factor in certain systemic conditions such as atherosclerotic heart diseases and aspiration pneumonia [92]. For a long time, the presence of these bacteria was thought to be the link between periodontal diseases and systemic diseases.

Liu et al. isolated outer membrane vesicles (OMVs) from *Fusobacterium nucleatum* [93] cultures and purified these using gradient centrifugation [94]. Through nano LC-MS/MS analysis, they identified 98 proteins within OMVs [94], of which 60 were predicted to localize to the outer membrane or periplasm via signal peptide-driven translocation. Of the aforementioned localized proteins, six autotransporter proteins (the majority of protein mass of OMVs) were associated with defective type V secretion systems [94]. In addition, other putative virulence factor proteins with functional domains, including FadA, MORN2 and YadA-like domain, were by in silico analysis. Identified with multiple exposed epitope sites as determined, Liu et al. concluded that the non-replicative OMVs of *F. nucleatum* contain multiple antigenic virulence factors that may play important roles in the design and development of vaccines against *F. nucleatum* [94].

Savickiene et al. isolated proanthocyanidin fraction from *Pelargonium sidoides* root extract (PSRE) and analyzed its composition [95]. The antibacterial efficiency of proanthocyanidin was compared to that of PSRE as a whole. Two bacterial strains were selected for assays of PSRE and proanthocyanidin antibacterial efficacy, and also to investigate their selectivity. Savickiene et al. used the Acquity UPLC system (Waters, Milford, CT, USA) equipped with a Quattro micro triple quadrupole tandem mass spectrometer (Waters) to determine the mean degree of polymerization [95]. An electrospray ionization source in negative mode was used to obtain MS/MS data in that study [95]. To present a detailed view of the proanthocyanidin composition, they used the Acquity UPLC to analyze PSRE and proanthocyanidin aliquots with MS/MS [95]. In this study by Savickiene et al., the results revealed that proanthocyanidins had a significantly stronger antioxidant capacity compared to the root extract and exhibited a unique antibacterial action profile that selectively targets Gram-negative periodontal and peri-implant pathogenic strains, such as *P.gingivalis*, while preserving the viability of beneficial oral commensal *Streptococcus salivarius* [95].

Recently, the MALDI Biotyper (Bruker Daltonics GmbH, Leipzig, Germany) was developed as a new system for bacterial identification. With the MALDI Biotyper, the traditional method of MALDI-TOF MS is used in combination with database software [96]. Collecting mass spectra of bacteria and comparing these to data in a database enabled rapid identification of bacteria. MALDI-TOF MS can be used to definitively identify, through mass spectra analysis, a large number of microbial species, including protein profiles of *Prevotella intermedia* and *Prevotella nigrescens* [97,98].

Rams et al. used MALDI-TOF MS to assess the accuracy of the phenotypic scheme for recognition of periodontal *P. intermedia* and *P. nigrescens* group isolates [99]. This study isolated 84 subgingival isolates that were phenotypically identified as belonging to these species [99], and of these isolates, 71.4% were confirmed as either *P. intermedia* or *P. nigrescens* with a log score 1.7 or higher by MALDI-TOF MS analysis, using Bruker Daltonics MALDI Biotyper analytic software for mass spectra [99]. The phenotypic scheme used correctly identified most of the group isolates [99], and thus rapid phenotypic identification of cultivable *P. gingivalis* in human subgingival biofilm specimens was found to be 100% accurate with MALDI-TOF mass spectrometry. These results provide validation for the continued use of *P. gingivalis* research data based on this species identification methodology [99].

Noëla et al. described the first case of fatal thoracic empyema caused by *Campylobacter rectus*, which was identified successfully using MALDI-TOF MS [100]. *Campylobacter rectus* is a rare anaerobic pathogen. In addition to being the first reported case of fatal *C. rectus*-associated thoracic empyema, it was only the second reported case in which MALDI-TOF MS was used to successfully identify the pathogen [100]. In this case report, *Campylobacter rectus* seems highly susceptible to most anaerobic-targeting antibiotics and poor dental hygiene appears to be the leading risk factor for *C. rectus* systemic infections [100].

In a study by Lönn et al., plasma lipoproteins were isolated from whole blood treated with wild-type and gingipain-mutant (lacking either the Rgp or Kgp gingipains) [101]. *Porphyromonas gingivalis* was studied using two-dimensional gel electrophoresis, followed by MALDI-TOF MS analysis. The main finding was that *Porphyromonas gingivalis* exerted substantial proteolytic effects on lipoproteins. Additionally, the Rgp gingipains were responsible for producing 2 apoE fragments, as well as 2 apoB-100 fragments, in LDL; the Kgp gingipain also produced an unidentified fragment in high-density lipoproteins. Moreover, *Porphyromonas gingivalis* and its different gingipain variants induced reactive oxygen species ROS and consumed antioxidants. Both Rgp and Kgp gingipains were involved in inducing lipid peroxidation [101]. Lönn et al. concluded that *P. gingivalis* had the potential to alter the concentrations of lipoproteins in blood. Together, these findings may represent a crucial link between periodontal and cardiovascular disease. Lönn et al. also suggested that periodontal bacteria, such as *P. gingivalis*, modify vascular low-density, very-low-density, and high-density lipoprotein to an atherogenic form [101]. Therefore, modified lipoproteins may serve as the key link between periodontal disease and atherosclerosis.

Rams et al. [102] used MALDI-TOF MS to assess the accuracy of the rapid phenotypic identification scheme used to facilitate detection of cultivable *P. gingivalis* in human subgingival plaque biofilms [102]. Each presumptive *P. gingivalis* clinical isolate was evaluated together with a panel of other human subgingival bacterial species, including clinical periodontal isolates of *P. intermedia*, *P. nigrescens, Prevotella melaninogenica*, *P. denticola*, *Eubacterium brachy*, *F. nucleatum*, and *Parvimonas micra*, and was subjected to MALDI-TOF MS analysis. *P. gingivalis* served as the reference for bacterial protein profiles (Bruker Daltonics, Billerica, MA, USA). Phenotypic identification of culturable *P. gingivalis* was found to be 100% accurate [102], as all 314 (100%) presumptive *P. gingivalis* subgingival isolates were confirmed as *P. gingivalis* using MALDI-TOF mass spectrometry analysis (Cohen’s kappa coefficient ¼ 1.0). MALDI-TOF mass spectrometry log scores between 1.7 and 1.9, and 2.0 were found for 92 (29.3%) and 222 (70.7%), of the presumptive *P. gingivalis* clinical isolates, respectively [102]. No other tested bacterial species was identified as *P. gingivalis* by MALDI-TOF mass spectrometry [102].

Van der Cruyssen et al. reported that stereotactic drainage and MALDI-TOF MS of the pus (Bruker Daltonics MALDI Biotyper, Bremen, Germany) revealed that *P. gingivalis* was the sole causative bacterium [103]. Intraoral inspection revealed that partial dentition was affected by periodontitis. *P. gingivalis* is a rare but potentially life-threatening anaerobe that can cause intracerebral abscesses. This study detailed the sixth reported case of *P. gingivalis* causing an intracranial abscess and the third case of a true intracerebral parenchymal abscess caused by this bacterium [103].

Recently, results of a single-blind, prospective, randomized, controlled clinical trial revealed the effectiveness of ozone gas or sodium hypochlorite/chlorhexidine disinfection protocol for root canal treatment of apical periodontitis. Kist et al. obtained microbial samples after preparing the access cavity, followed by chemo-mechanical treatment. Then, they performed inter-appointment dressing by sterile paper points [104]. Microbial identification was performed with MALDI-TOF MS and 16S ribosomal RNA gene sequencing. The bacterial genera most commonly found were *Streptococcus*, *Parvimonas*, and *Prevotella* [104]. There were no significant differences between success rates. Conversely, the differences in the decreases in (periapical index) PAI values and apical lesion sizes were also insignificant after 6 and 12 months. The bacterial reduction showed no significant differences between groups after chemo-mechanical treatment and after inter-appointment dressing. Moreover, the most commonly found bacterial genera were *Streptococcus* spp., *Parvimonas* spp. and *Prevotella* spp. [104]. They concluded that the use of ozone gas and sodium hypochlorite/chlorhexidine protocols had no effect on bacterial reduction in the sampled areas of the root canals [104].

Stîngu et al. evaluated the use and value of MALDI-TOF MS analysis for rapid identification of different species of anaerobic bacteria cultivated from the subgingival biofilm [105]. The mass spectra of the *P. intermedia* strains identified with MALDI were clustered together but were separate from the spectra of *P. nigrescens*. The reference strains of the anaerobic bacteria used showed peaks between m/z 2000 and up to about m/z 13,000 as characteristic MALDI-TOF-MS spectra. The similarity of spectra produced by strains of a single genus could be recognized visually. Obvious differences between spectra produced by strains of different species were also easily noticed. This spectra of the *Prevotella intermedia* strains identified with MALDI clustered together and clustered separately from the spectra of *Prevotella nigrescens*, suggesting that MALDI-TOF-MS is an accurate method capable of separating these two species [105]. The quality of clustering was characterized by calculating an inconsistency coefficient [105].

Oscarsson et al. reported that MALDI-TOF MS and immunoblotting revealed that both bacterial phenotypes, including the biofilm and planktonic forms of *Aggregatibacter actinomycetemcomitans*, released significant amounts of GroEL-like protein in free-soluble form [106]. Conversely, the immunomodulatory toxins, cytolethal distending toxin and leukotoxin, and peptidoglycan-associated lipoprotein appeared to be less important; this was evident by studying the strain D7S *cdt*/*ltx* double, and *pal* single mutants [106]. In addition to *A. actinomycetemcomitans,* a non-oral species, *Escherichia coli* strain IHE3034, tested in the same *ex vivo* model, also released free-soluble surface material with proinflammatory activity [106]. Their data suggests that release of surface components from live bacterial cells could constitute a mechanism for systemic stimulation and be particularly important in chronic localized infections, such as periodontitis [106].

### 6.3. Virus Infection

The increased prevalence and severity of chronic periodontal disease in patients with human immunodeficiency virus (HIV) infection suggest that HIV infection predisposes patients to chronic periodontitis [107]. 

In recent years, MALDI-TOF-MS and LC-MS/MS have both been used widely to analyze the composition of purified virions. As such, previously unidentified components of viral particles have been discovered and reported [108].

Gao et al. performed a comprehensive meta-analysis of the literature to confirm that the Epstein–Barr virus was associated with periodontal diseases [107]. The results suggest that detection of the Epstein–Barr virus is correlated with increased risk of periodontal diseases [107]. Meta-analysis of 21 studies including 995 patients with periodontal diseases and 564 healthy people suggests that EBV is associated with increased risks of periodontitis including CP and AgP [107]. This relationship exists in Asians, Europeans, and Americans. SgP and tissue are available for detecting EBV in patients with periodontitis. However, because of a lack of sufficient evidence, detecting EBV in GCF samples still remains uncertain. They concluded that that detecting EBV in samples from ≥5 (6) mm sites of periodontal pockets is more sensitive than from 3-mm sites [107].

Recently, proteome analysis has been used to characterize virus–host interplay [109]; methods including integrative MS, antibody-based affinity purification of protein complexes, cross-linking, and protein array techniques have all been used to elucidate complex networks of virus–host protein associations during infection with both RNA and DNA viruses [109].

To gain insight into the biological processes that may be disrupted in response to viral infection and to illustrate global similarities and differences between viruses, Reyes et al. [110] performed comparative gene ontology (GO) enrichment analysis of host proteins for each host–virus pair. Reyes et al observed that gene ontologies characterized by “enriched” proteins revealed distinctions between harnessed and differentiated viruses that undergo nuclear replication (Ad5 and HSV-1) from those that undergo cytoplasm replication (VACV) [110].

Proteomic analysis has become the leading approach for large-scale analysis of complex biological samples. Its importance in clinical applications has further increased owing to the development of high-performance instruments, which allow for the identification of disease-specific biomarkers and rapid protein profiling of the analyzed samples. Proteomics data have driven the development of specific bioinformatics tools that can assist researchers during the discovery process. Specific biomarkers also represent key features for improving the methods of diagnosis and prognosis. The application of the Western blot method alone for validation lacked conclusive evidence. The development of specific bioinformatics tools that can be used for proteome analysis will be required in order to assist researchers in the effort to improve our understanding of global biological systems.

## 7. Outlook for the Future

At present, the search for molecular events in the pathogenesis of viral periodontal disease continues, with the use of proteome analysis (Table 2). However, the traditional approach of relying on expression proteomics is limited by the fact that most proteins acquire their native function only after undergoing some post-translational modification. Thus, post-translational modification of proteins has also been a subject of research. Many types of post-translational modification of proteins are known means of diversifying protein functions. In research, not only the level of expression of proteins but also post-translational modifications must be considered. Recent technological innovations have enabled comprehensive searches of post-translational modifications of proteins (modificomics). Representative examples include phosphorylation, glycosylation, acetylation, and methylation. Because of their comprehensive nature, proteomics and modificomics are expected to help identify numerous proteins that seem to be related to periodontal disease; at the same time, the mechanism of their action is expected to be elucidated, along with identification of proteins that may be specific markers of both periodontal disease and molecular events in the pathogenesis of viral periodontal disease.

Other techniques have been used to identify relevant genes and inflammatory markers. Studies performed to date have evaluated the microbiota of EPL using culture [31,111,112] and have employed targeted molecular techniques including polymerase chain reaction (PCR) [30,31,36,38], real-time PCR [32] and checkerboard DNA–DNA hybridization [107,113], open-ended molecular techniques (next generation sequencing; NGS) [37] and denaturing gradient gel electrophoresis (DGGE) and/or gene cloning and sequencing [36,38]. Overall, these studies revealed significant similarities between the microbiota found in the root canals and those in periodontal pockets. Furthermore, developments in glycan microarray technologies have led to targeted identification of glycan-based receptors [113]. The mapping of cell surface interactions of the diverse types of adenovirus has implications for cells, tissues, and other structures, as well as for drug development. Adenovirus interactions with glycan- and protein-based receptors, as well as glycomic and proteomic strategies, have been used in an attempt to identify virus receptors and attachment factors.

Proteomic approaches with the use of MS can quantify the post-translational modifications of host and virus proteins during the course of infection [114]. Most studies of virus-mediated post-translational modifications use either stable isotope labeling with amino acids in cell culture (SILAC) or isobaric labeling; however, label-free quantitative proteomics is becoming more popular due to the improved sensitivity of instruments.

## 8. Conclusions

MALDI-TOF MS and LC-MS/MS are now essential tools for discovering molecular events in the oral pathogenesis of viral infections, particularly in periodontitis. Their contribution to detection of susceptibility, however, is still limited. For this purpose, LC coupled with MS/MS should be more useful and will play significant roles in this field of research in the future. Recent advancements in the field of cellular biology have improved the possibility of new and more effective treatments for many illnesses that affect human beings and will help clarify the poorly understood mechanisms of these molecular events.

## Figures and Tables

**Table 1 ijms-21-05184-t001:** Applications of MALDI-TOF MS and LC-MS/MS in laboratory medicine.

**MALDI-TOF MS**
• Clinical microbiology
• Imaging MS
**LC-MS/MS**
• Inborn errors of metabolism
• Therapeutic drug monitoring
• Toxicology
• Endocrinology
• Targeted metabolomics, peptidomics, and proteomics
• Clinical microbiology

MS, mass spectrometry; MALDI-TOF MS, matrix-assisted laser desorption/ionization time-of-flight mass spectrometry; LC-MS/MS, liquid chromatography with tandem mass spectrometry.

**Table 2 ijms-21-05184-t002:** Proteome Analyses of Molecular Events in Oral Pathogenesis of Virus Infection in GCF, Saliva, and Other Oral Components in Periodontal Disease.

Study	Methodology	Objects	Essence of a Discourse or Summary of Results
Savickiene et al. [95]	Ultra-performance liquid chromatography-electrospray tandem mass spectrometryultra-performance liquid chromatography-electrospray tandem mass spectrometry (UPLC-ESI-MS) was used to analyze plant-derived proanthocyanidins (PACN) to help determine whether they showed an antibacterial effect on periodontopathogenic bacteria. PACN were purified from pelargonium sidoides DC root extracts (PSRE) using acid/n-Butanol hydrolysis. PACN and PSRE exhibited antibacterial activity against Gram-negative periodontal and peri-implant pathogenic strains like *P. gingivalis* but maintained viability of commensal bacteria like *S. salivarius*.	Effect of PSRE and PACN on *Porphyromonas gingivalis* and *Streptococcus Salivarius*	The results suggested that proanthocyanidins had significantly stronger antioxidant capacity than did the root extract and exhibited unique antibacterial action profile that selectively targets Gram-negative periodontal and peri-implant pathogenic strains, such as *P. gingivalis*, while preserving the viability of beneficial oral commensal strains, such as *S. salivarius*.
Rams et al. [99]	Matrix-assisted laser desorption/ionization time-of-flight mass spectrometry (MALDI-TOF-MS) was used to identify *P. intermedia/nigrescens* group from clinical isolates. Using a >1.7 log score agreement threshold, 71.4% of the presumptive isolates from 23 adults with chronic periodontitis were identified in the assay.	*Porphyromonas intermedia* and *Prevotella nigrescens* group	MALDI-TOF MS was used to assess accuracy of the phenotypic scheme for recognition of periodontal P. intermedia and P. nigrescens group isolates. Of 84 subgingival isolates that were phenotypically identified as belonging to the two species, 71.4% were confirmed as either *P. intermedia* or *P. nigrescens,* with a log score of 1.7 or more. The phenotypic scheme was used correctly and identified most of the group isolates. Therefore, rapid phenotypic identification of cultivable *P. gingivalis* in human subgingival biofilm specimens was found to be 100% accurate with MALDI-TOF mass spectrometry. These results validate the continued use of *P. gingivalis* research data that are based on this method of species identification.
Noëla et al. [100]	MALDI-TOF-MS analysis was used to identify *Campylobacter rectus* in a patient with thoracic empyema.	*Campylobacter rectus*	This was the first case of fatal thoracic empyema caused by *C. rectus,* which was identified by MALDI-TOF MS. *C. rectus* is a rare anaerobic pathogen. This case was also only the second reported case in which MALDI-TOF MS was used to successfully identify the pathogen. *C. rectus* seems highly susceptible to most anaerobic-targeting antibiotics, and poor dental hygiene appears to be the leading risk factor for systemic *C. rectus* infections.
Lönn et al. [101]	MALDI-TOF-MS following 2D PAGE was used to examine alterations in plasma lipoproteins induced by the periodontopathic bacterium *P. gingivalis* in vitro. Plasma lipoproteins isolated from whole blood were examined using MALDI-TOF-MS analysis. *P. gingivalis* and its gingipain variants induced lipid peroxidation, induced lipid peroxidation, as measured using thiobarbituric acid-reactive substances; lipoprotein proteolysis, as measured by MS; and ROS induction and antioxidant consumptions, as assessed using antioxidant assay kits and lumiaggregometry, respectively.	*P. gingivalis,* lipoproteins	The main finding was that *P. gingivalis* exerted a substantial proteolytic effect on lipoproteins. The Rgp gingipains were responsible for producing two apoE fragments and two apoB-100 fragments in LDL. The Kgp gingipain produced an unidentified fragment in HDL. Moreover, *P. gingivalis* and its different gingipain variants induced ROS and consumed antioxidants. Both Rgp and Kgp gingipains were involved in inducing lipid peroxidation. The authors concluded that *P. gingivalis* had the potential to alter the concentrations of lipoproteins in blood. These findings may represent a crucial link between periodontal and cardiovascular disease. Periodontal bacteria, such as *P. gingivalis,* may modify vascular LDL, very-low-density lipoprotein, and HDL into an atherogenic form.
Rams et al. [102]	MALDI-TOF MS was used to specifically identify *P. gingivalis* from a mixture of other human subgingival bacteria species. Using a >1.7 log score agreement threshold, presumptive *P. gingivalis* isolates from 38 adults with chronic periodontitis were identified with 100% accuracy in the assay.	*P. gingivalis* in human subgingival plaque biofilms	Phenotypic identification of culturable *P. gingivalis* was found to be 100% accurate. All 314 (100%) presumptive *P. gingivalis* subgingival isolates were confirmed as *P. gingivalis* with MALDI-TOF MS analysis (Cohen’s kappa coefficient = ¼ 1.0). MALDI-TOF MS log scores between 1.7 and 1.9 and of 2.0 were found for 92 (29.3%) and 222 (70.7%), respectively, of the presumptive *P. gingivalis* clinical isolates. No other tested bacterial species was identified as *P. gingivalis* by MALDI-TOF MS.
Van der Cruyssen et al. [103]	MALDI-TOF MS analysis using a Bruker Daltonics MALDI Biotyper was used to identify *P. gingivalis* in a patient with a cerebral abscess.	*P. gingivalis*	Stereotactic drainage and MALDI-TOF MS of the pus (Bruker Daltonics MALDI Biotyper) revealed that *P. gingivalis* was the sole causative bacterium. Intraoral inspection revealed that the partial dentition was affected by periodontitis. *P. gingivalis* is a rare but potentially life-threatening anaerobe that can cause intracerebral abscesses. This article was about the sixth reported case of intracranial abscess caused by *P. gingivalis* and the third case of a true intracerebral parenchymal abscess caused by *P. gingivalis.*
Kist et al. [104]	MALDI-TOF was used in a single-blinded prospective randomized controlled clinical trial to compare the efficacy of ozone gas versus sodium hypochlorite/chlorhexidine (NaOCl/CHX) in disinfecting root canals for treatment of apical periodontitis. Following cleansing with NaCl and EDTA, root canals were treated with ozone gas or NaOCl/CHX. Microbial samples were taken at multiple time points during treatment and analyzed by MALDI-TOF-MS and 16S-rRNA gene. No significant differences were observed between success rates.	*Streptococcus, Parvimonas,* and *Prevotella-associated*	The bacterial genera most commonly found were *Streptococcus, Parvimonas,* and *Prevotella.* There were no significant differences between success rates. Conversely, the differences between the decreases in periapical index values and apical lesion sizes were also insignificant after 6 and 12 months. The bacterial reduction showed no significant differences between groups after chemomechanical treatment and after interappointment dressing. In addition, the use of ozone gas and sodium hypochlorite/chlorhexidine protocols had no effect on bacterial reduction in the sampled areas of the root canals.
Stîngu et al. [105]	MALDI-TOF-MS was used to identify anaerobic clinical isolates from patients with periodontal disease. 84 strains that were previously genotypically identified by sequence analysis of 16S ribosomal RNA were analyzed using MALDI-TOF-MS. Spectra of *P. intermedia* strains clustered separately from the spectra of *Prevotella nigrescens*, indicating this method can accurately distinguish between these species.	*Anaerobic bacteria,* subgingival biofilm	The mass spectra of the *P. intermedia* strains identified with MALDI-TOF MS were clustered together but were separate from the spectra of *P. nigrescens.* The reference strains of anaerobic bacteria used showed peaks between m/z 2000 and approximately m/z 13,000, characteristic of MALDI-TOF-MS spectra. The similarity in spectra produced by strains of a single genus could be recognized visually. Obvious differences between spectra produced by strains of different species were also easily noticed. The spectra of the *P. intermedia* strains identified with MALDI-TOF MS clustered together and clustered separately from those of the *P. nigrescens* strains; these data show that MALDI-TOF-MS is an accurate method capable of differentiating these two species. To characterize the quality of clustering, an inconsistency coefficient was calculated.
Oscarsson et al. [106]	MALDI-TOF-MS was used in an *ex vivo* cell culture insert model to determine the factors released by *A. actinomycetemcomitans* strain D7S, which is linked to periodontitis. Whole blood samples were stimulated with planktonic and biofilm forms of *A. actinomycetemcomitans* and proinflammatory cytokine production was identified using cytokine antibody arrays/immunoassays. SDS-PAGE, MALDI-TOF mass spectrometry, and quantitative real-time PCR analyses revealed that the release of GroEL-like protein in free-soluble form induced the inflammatory response.	Biofilm and planktonic forms of *Aggregatibacter actinomycetemcomitans,* GroEL-like protein	MALDI-TOF MS and immunoblotting revealed that both the biofilm and planktonic forms of *A. actinomycetemcomitans* released significant amounts of GroEL-like protein in free soluble form. Conversely, the immunomodulatory toxins cytolethal distending toxin and leukotoxin, as well as peptidoglycan-associated lipoprotein, appeared to be less important, as evidenced by study of the strain D7S *cdt/ltx* double and *pal* single mutants. In addition to *A. actinomycetemcomitans,* a nonoral species, *Escherichia coli* strain IHE3034, tested in the same ex vivo model, also released free-soluble surface material with proinflammatory activity. The results suggest that the release of surface components from live bacterial cells could constitute a physiological mechanism for systemic stimulation and be particularly important in chronic localized infections, such as periodontitis.
Bostanci et al. [90]	label-free liquid chromatography mass spectrometry in data-independent analysis mode (LC/MS-E) was used to identify and quantify multiple proteins simultaneously in gingival crevicular fluid exudatome from healthy and periodontally diseases patients.	154 proteins of human, bacterial, and viral origin from GCF and viral proteins (such as herpes virus protein 2)	The proportion of viral protein was higher in GCF samples from patients with periodontal disease than in healthy patients. Higher levels of viral proteins (such as herpesvirus protein 2) in the diseased samples corroborated evidence from previous reports that confirm the involvement of viral infection in the pathogenesis of periodontal disease.
Gao et al. [107]	A comprehensive meta-analysis from a systematic literature search using key terms "EBV" and "periodontitis OR peridontal disease" to assess the relationship between Epstein-Barr Virus and periodontitis. Publications were included if they were case-control studies, estimated the association between periodontal diseases and EBV, extracted samples using surgery, paper point, curette, paper strip, or biopsy, patients were systemically healthy, and sample sizes, odds ratios, and 95% confidence intervals were included. Publications were excluded if no useful data could be obtained or periodontitis or periodontitis-similar diseases were not diagnosed. The odds ratios with 95% confidence intervals were used to assess the strength of associations. They found a correlation between EBV and an increased risk of periodontal disease.	*Epstein–Barr virus*	The results indicated that the detection of EBV is correlated with increased risk for periodontal diseases. The results also suggest that EBV is associated with increased risks of periodontitis, including chronic and aggressive periodontitis. This relationship exists in Asians, Europeans, and Americans. Subgingival plaque and tissues were available for detecting EBV in patients of periodontitis. However, because of a lack of sufficient evidence, detecting EBV in GCF sample still remains uncertain. In conclusion, the results suggest that samples from ≥5-mm sites of periodontal pockets are more sensitive for detecting EBV than those from ≥3-mm sites.
Liu et al. [94]	Nano LC/MS/MS was used to determine if outer membrane vesicles (OMVs) from *Fusobacterium nucleatum* contained antigenic proteins that could be used in vaccine development. Proteins contained within the OMVs were identified using the MS analysis, and epitope sites in the resulting proteins were determined by *in silico* analysis.	Outer membrane vesicles (OMVs) from *Fusobacterium nucleatum*	Of the 60 proteins predicted to localize to the outer membrane or periplasm, proteins, six autotransporter proteins (the majority of protein mass of OMVs) were associated with defective type V secretion systems. In addition, other putative virulence factor proteins with functional domains, including FadA, MORN2, and YadA-like domain, were found by in silico analysis. They were identified with multiple exposed epitope sites. The nonreplicative OMVs of *F. nucleatum* contain multiple antigenic virulence factors that may play an important role in the design and development of vaccines against *F. nucleatum* infection.
Ngo et al. [85]	MALDI-MS and LC MALDI-MS, LC-MS of in-gel digested proteins in RP-HPLC, followed by MALDI-TOF/TOF MS/MS for the identification of GCF peptides.	GCF	With the techniques described, 33 peptides were identified; these corresponded to cleavage products that had not previously been reported in GCF; in addition, 66 proteins, including 43 newly discovered proteins, were identified in GCF; these findings represented the most comprehensive proteomic information about GCF to date. Use of RP-HPLC followed by MALDI-TOF/TOF MS/MS was described a new method of identifying GCF peptides.
Ngo et al. [86]	Gingival crevicular fluid analyzed with MALDI-TOF MS.	GCF	A genetic algorithm was used to create a model based on pattern analysis to predict sites of underlying attachment loss. The clinical indices pocket depth, modified gingival index, plaque levels, and bleeding on probing were poor discriminators of mass spectra of GCF. Models generated from the GCF mass spectra could predict attachment loss at a site with high specificity, 97% recognition capability, and 67% cross-validation.
Chaiyarit et al. [87]	MALDI-TOF/TOF MS for analyzing specific patterns of mass signals of low-molecular-weight proteins in saliva from patients with different oral diseases.	Whole saliva samples from healthy patients and those with oral diseases, including oral cancer, oral lichen planus, and chronic periodontitis.	The percentages of mass signals at 5592.26 and 8301.46 Da from oral cancer were significantly higher than those from other diseases (*p* = 0.002 and *p* = 0.030, respectively). In oral lichen planus, the percentages of mass signals at 12,964.55 and 13,279.08 Da were significantly higher than those of other groups (*p*s < 0.001). In chronic periodontitis, the percentages of mass signals at 5835.73 and 9801.83 Da were significantly lower than those of other groups (*p* = 0.003 and *p* = 0.005, respectively).
Antezack et al. [88]	MALDI-TOF analysis could accurately classify these three types of samples according to periodontal state.	Saliva, GCF, and dental plaque.	Rapid routine and blinded MALDI-TOF analysis could accurately classify these three types of samples according to periodontal state (healthy and diseased). Diagnostic tests based on protein profiles in saliva, GCF, and dental plaque were developed for the first time. Among 67 periodontitis and 74 healthy controls, the decision trees enabled diagnosis of periodontitis with 70.3% sensitivity (± 0.211) and 77.8% specificity (± 0.165) for saliva, with 79.6% sensitivity (± 0.188) and 75.7% specificity (± 0.195) for GCF, and with 72.1% sensitivity (± 0.202) and 72.2% specificity (± 0.195) for dental plaque. Both the sensitivity and specificity of the tests were improved to 100% (95% CIs = 0.91–1 and 0.92–1, respectively) when two samples were tested.
Tang et al. [89]	Samples from 17 patients with gingivitis and 16 periodontally healthy persons as controls were analyzed with MALDI-TOF MS. Nano liquid chromatography tandem mass spectrometry (nano-LC/ESI-MS/MS) was performed to identify possible proteins.	Whole saliva, GCF, and serum samples	Levels of most of the differentially expressed peptides were increased in participants with chronic periodontitis and gingivitis, in comparison with healthy controls. Cluster analysis showed good differentiation between patients with chronic periodontitis and healthy controls. Most AUCs for differentially expressed peptides were >0.7, whereas some peptides from GCF and serum exhibited AUCs as high as 0.9–1.0.

AUC, area under the curve; CI, confidence interval; GCF, gingival crevicular fluid; HDL, high-density lipoprotein; LC-MS, liquid chromatography–mass spectrometry; LDL, low-density lipoprotein; ROS, reactive oxygen species; RP-HPLC, reversed-phase high-performance liquid chromatography.

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
