# Peer review of "Proteome Analysis of Molecular Events in Oral Pathogenesis and Virus: A Review with a Particular Focus on Periodontitis"

_ijms, 2020, doi:10.3390/ijms21155184_

Round 1
Reviewer 1 Report
I think this review brings a relevant discussion about the identification of proteomes in periodontitis. I do not have big criticism in this review. However, I would suggest the inclusion of a brief description about where we are now in terms of other techniques despite proteome analysis (genes, inflammatory markers….) Also, enhance the “outlook for the future” section, adding the possible clinical applications using this technology.
Author Response
Point-by-point response to reviewers’ comments
Manuscript ID: ijms-780206
Type of manuscript: Review
Title: Proteome Analysis of Molecular Events in Oral Pathogenesis and Virus: A Review with a Particular Focus on Periodontitis
Thank you very much for your valuable comments and suggestions regarding our manuscript. The manuscript has been revised completely according to the suggestions provided. The changes made are as indicated below in the point-by-point responses.
Reviewers' comments:
Reviewer 1
I think this review brings a relevant discussion about the identification of proteomes in periodontitis. I do not have big criticism in this review. However, I would suggest the inclusion of a brief description about where we are now in terms of other techniques despite proteome analysis (genes, inflammatory markers….) Also, enhance the “outlook for the future” section, adding the possible clinical applications using this technology.
Response:
I am grateful for this thoughtful suggestion. I have added information from the literature regarding critical genes and inflammatory markers that have emerged from the use of a wide variety of techniques and methodologies. These additions are included in the final section entitled “Outlook for the Future.” (Lines 373-378, page 9)
Reviewer 2 Report
Dear Author,
Your review has high standard aims. however, the introduction and the patogenesis on periodontitis appear unclear and confused.
I suggest to rewrite in a more schematic and clear way, maybe relying on the recent classification of periodontal disease released in 2018.
also the organization of the paragraphs are really messy.
at some point you name the laser, and this is ok, but I think that re-ordering the paragraph would make the article readable and comprehansible.
also the virus theme. It deserves attention indeed. therefore, please dedicate paragraphs to them and to their diagnosis.
Author Response
Point-by-point response to reviewers’ comments
Manuscript ID: ijms-780206
Type of manuscript: Review
Title: Proteome Analysis of Molecular Events in Oral Pathogenesis and Virus: A Review with a Particular Focus on Periodontitis
Thank you very much for your valuable comments and suggestions regarding our manuscript. The manuscript has been revised completely according to the suggestions provided. The changes made are as indicated below in the point-by-point responses.
Reviewer 2
Your review has high standard aims. however, the introduction and the patogenesis on periodontitis appear unclear and confused.
I suggest to rewrite in a more schematic and clear way, maybe relying on the recent classification of periodontal disease released in 2018.
also the organization of the paragraphs are really messy.
at some point you name the laser, and this is ok, but I think that re-ordering the paragraph would make the article readable and comprehansible.
also the virus theme. It deserves attention indeed. therefore, please dedicate paragraphs to them and to their diagnosis.
Response:
I thank the Reviewer for this thoughtful suggestion. The manuscript has been reorganized and now includes a focus on the 2018 classifications of periodontal disease. (Lines 66-91, page 2-3) (Lines 47-50, page 2)
The paragraphs have also been re-ordered so that the readers can comprehend the points made more clearly.
Reviewer 3 Report
This article addresses several topics potentially interesting but not really well described and organized. The form of narrative review is misleading, as it's not clear the real focus of this review, starting from the title that do not indicate the main aim of this work. The methodology is missing or badly described. The discussion is wide but not focused on specific points. Generally, I do not recommend acceptance.
Author Response
Point-by-point response to reviewers’ comments
Manuscript ID: ijms-780206
Type of manuscript: Review
Title: Proteome Analysis of Molecular Events in Oral Pathogenesis and Virus: A Review with a Particular Focus on Periodontitis
Thank you very much for your valuable comments and suggestions regarding our manuscript. The manuscript has been revised completely according to the suggestions provided. The changes made are as indicated below in the point-by-point responses.
Reviewer 3
This article addresses several topics potentially interesting but not really well described and organized. The form of narrative review is misleading, as it's not clear the real focus of this review, starting from the title that do not indicate the main aim of this work. The methodology is missing or badly described. The discussion is wide but not focused on specific points.
Response:
I thank the Reviewer for bringing up these important points. I have provided additional literature and descriptions of the methodologies involved in MALDI-TOF MS and LC-MS; see Section 4 (Lines 191-203, page 5) and Table 2. I also reorganized the paragraphs in order to clarify the text.
Round 2
Reviewer 2 Report
Manuscript was highly improved.
I still find the abstract has to be modified and adapted to the content of the manuscript
Author Response
Point-by-point response to reviewers’ comments
Manuscript ID: ijms-780206
Type of manuscript: Review
Title: Proteome Analysis of Molecular Events in Oral Pathogenesis and Virus: A Review with a Particular Focus on Periodontitis
Thank you very much for your valuable comments and suggestions regarding our manuscript. The manuscript has been revised completely according to the suggestions provided. The changes made are as indicated below in the point-by-point responses
Reviewers' comments:
Reviewer 2
Manuscript was highly improved.
I still find the abstract has to be modified and adapted to the content of the manuscript.
Response:
Thank you for this thoughtful suggestion. The abstract has been modified to better reflect the contents of the manuscript.
Reviewer 3 Report
Unfortunately, the changes made by the author did not improved the overall quality of this paper. It is rich of information, however, it is misleading and it addresses too many focuses, often poorly linked within the text. One table cannot be correctly read. The only one readable misses of information on the results of the studies reported: methods and aim are unuseful to readers if you do not report the results achieved. The conclusions seems a technical note on the usefulness of some technologies instead to read something on novel diagnostic pathways or some novel therapeutic approach. I suggest author to critically rebuild his paper, reducing the text to 1-2 specific points of interest, focussing it on them in a "debate" article, or in a "technical note". The form of "review" could be also used if author change the methods (nothing has been reported on the criteria used for this review). At this point, I suggest rejection encouraging resubmission.
Author Response
Point-by-point response to reviewers’ comments
Manuscript ID: ijms-780206
Type of manuscript: Review
Title: Proteome Analysis of Molecular Events in Oral Pathogenesis and Virus: A Review with a Particular Focus on Periodontitis
Thank you very much for your valuable comments and suggestions regarding our manuscript. The manuscript has been revised completely according to the suggestions provided. The changes made are as indicated below in the point-by-point responses.
Reviewers' comments:
Reviewer 3
Unfortunately, the changes made by the author did not improved the overall quality of this paper. It is rich of information, however, it is misleading and it addresses too many focuses, often poorly linked within the text. One table cannot be correctly read. The only one readable misses of information on the results of the studies reported: methods and aim are unuseful to readers if you do not report the results achieved. The conclusions seems a technical note on the usefulness of some technologies instead to read something on novel diagnostic pathways or some novel therapeutic approach.
Response:
We thank the Reviewer for raising these important points. We provided additional literature, as well as the Essence of discourse or summary of results involved. Please see Section 5 and Table 2
Round 3
Reviewer 3 Report
The topic of this article entitled “Proteome Analysis of Molecular Events in Oral Pathogenesis and Virus: A Review with a Particular Focus on Periodontitis.” is aimed to discuss proteome analysis of molecular events in the pathogenesis of oral diseases and viruses, with a particular focus on periodontitis. Authors have well revised several issues; however, I ask authors to add some key concepts. Authors have reported several important topics related to tissue pathogenesis, healing and repairing; however, poor has been reported on the other resident stem cells, which can act as the immunomodulatory and pro-osteogenic activities in the local environment (Please, see and discuss: Commitment of Oral-Derived Stem Cells in Dental and Maxillofacial Applications. Dent J (Basel) 2018, 6, 72.) moreover, the function of MSCs in translational applications must be introduced as a general background (Please, see and discuss: Mesenchymal Stem Cells as Promoters, Enhancers, and Playmakers of the Translational Regenerative Medicine 2018. Stem Cells Int. 2018 Oct 30;2018:69274019). Authors have discussed about “Proteome Analysis of Molecular Events in Oral Pathogenesis”. In this landscape, to date, it’s well-known that still remain issues related to use of some in-vitro and in-vitro reparative protocols. In this light it’s important to briefly describe something about the “safe” in-vitro reparative models, working without any additive (e.g. BSA) to apply safely such protocols in human models (Please, see and discuss “Highly Efficient In Vitro Reparative Behaviour of Dental Pulp Stem Cells Cultured with Standardised Platelet Lysate Supplementation. Stem Cells Int. 2016;2016:7230987.”
Author Response
Point-by-point response to reviewers’ comments
Manuscript ID: ijms-780206
Type of manuscript: Review
Title: Proteome Analysis of Molecular Events in Oral Pathogenesis and Virus: A Review with a Particular Focus on Periodontitis
Thank you very much for your valuable comments and suggestions regarding our manuscript. The manuscript has been revised completely according to the suggestions provided. The changes made are as indicated below in the point-by-point responses.
Reviewers' comments:
Reviewer 3
The topic of this article entitled “Proteome Analysis of Molecular Events in Oral Pathogenesis and Virus: A Review with a Particular Focus on Periodontitis.” is aimed to discuss proteome analysis of molecular events in the pathogenesis of oral diseases and viruses, with a particular focus on periodontitis. Authors have well revised several issues; however, I ask authors to add some key concepts. Authors have reported several important topics related to tissue pathogenesis, healing and repairing; however, poor has been reported on the other resident stem cells, which can act as the immunomodulatory and pro-osteogenic activities in the local environment (Please, see and discuss: Commitment of Oral-Derived Stem Cells in Dental and Maxillofacial Applications. Dent J (Basel) 2018, 6, 72.) moreover, the function of MSCs in translational applications must be introduced as a general background (Please, see and discuss: Mesenchymal Stem Cells as Promoters, Enhancers, and Playmakers of the Translational Regenerative Medicine 2018. Stem Cells Int. 2018 Oct 30;2018:69274019). Authors have discussed about “Proteome Analysis of Molecular Events in Oral Pathogenesis”. In this landscape, to date, it’s well-known that still remain issues related to use of some in-vitro and in-vitro reparative protocols. In this light it’s important to briefly describe something about the “safe” in-vitro reparative models, working without any additive (e.g. BSA) to apply safely such protocols in human models (Please, see and discuss “Highly Efficient In Vitro Reparative Behaviour of Dental Pulp Stem Cells Cultured with Standardised Platelet Lysate Supplementation. Stem Cells Int. 2016;2016:7230987.”
Response:
We thank the Reviewer for raising these important concerns. We have provided additional literature regarding the involvement of Stem cells in periodontal tissue. Please check the new Section 4.
Round 4
Reviewer 3 Report
Article can be accepted
Author Response
Point-by-point response to Academic Editor comments
Manuscript ID: ijms-780206
Type of manuscript: Review
Title: Proteome Analysis of Molecular Events in Oral Pathogenesis and Virus: A Review with a Particular Focus on Periodontitis
Thank you very much for your valuable comments and suggestions regarding our manuscript. We have completely revised the manuscript in accordance with the suggestions provided. The changes made are as indicated below in the point-by-point responses.
Academic Editor comments:
- Overall, the objectives and conclusions of this study should be more clearly presented. How to take home a message regarding the proteomic analysis of molecular events in oral pathogenesis and viruses?
We thank you for carefully reading our manuscript. We have accordingly added the text in the revised manuscript (Introduction and Conclusion sections).
- Although this article is submitted in section "Etiopathogenesis of virus-associated oral diseases", and that "virus" appears as a main key word in the Title, the part devoted to the role of viruses in periodontitis is still superficial and poor (10 lines, pages 3 & 4). Most developments in this field are not cited. I believe that reading recent Jorgen Slots' review articles may help to improve this part.
We thank you for carefully reading our manuscript. We have accordingly added text in section 3 (Bacterial and Viral Pathogens Associated with Periodontal Disease).
- Why several pioneering works related to the use of MALDI TOF in periodontitis are missing? For example, Ngo LH et al 2010 & 2013; Chaiyarit P 2015; Tang H 2019; Angéline Antezack,2020, and many others...The author should explain the method followed to select their references (type of study, criteria, field...)?
We decided that references should indeed be up to-date. Therefore, we added section 6 and Table 2 and cite Ngo et al. (2010 and 2013), Chaiyarit et al. (2015), Tang et al. (2019), and Antezack et al. (2020).
- The presentation of periodontitis therapy is incomplete as initial non-surgical therapy for the removal of plaque and tartar by scaling and root planing is not mentioned ?
We thank you for your comment. We have added a paragraph about this in section 2 (Periodontal Disease).
- I agree that systemic diseases, such as diabetes, can aggravate periodontitis, but conversely periodontitis also has major direct consequences on dozens of systemic diseases (J.D. Beck, 2019). This point should be better explained.
We thank you for your comment. We have added several sentences about this to the first paragraph in section 2 (Periodontal Disease).
- Some parts are still confused, in particular part 5 (Proteomic analysis of molecular events ....... and viral infection) should be better written, in its current form it appears rather as appears as a listing of successive references with no links between them and no specific relevance...
In hopes of clarifying the discussion, we divided this section (now renumbered 6) into three subsections according to pathogenesis of virus, as follows:
- 6. Proteome Analysis of Molecular Events in Oral Pathogenesis of Virus in GCF, Saliva, and Other Oral Components in Periodontal Disease
6.1 GCF and saliva
6.2 Oral disease pathogenesis
6.3 Virus infection